# Effects of Concurrent Administration of BVDV Modified Live Viral Vaccine and RB51 on Immune Responses in Cattle

**DOI:** 10.3390/v17040553

**Published:** 2025-04-11

**Authors:** Lauren S. Crawford, Shollie Falkenberg, Steven C. Olsen, Paola M. Boggiatto

**Affiliations:** 1Infectious Bacterial Diseases Research Unit, National Animal Disease Center, Ames, IA 50010, USA; scruggs2307@gmail.com (L.S.C.); steven.olsen@usda.gov (S.C.O.); 2Oak Ridge Institute for Science and Education, Oak Ridge, TN 37830, USA; 3Immunobiology Graduate Program, Iowa State University, Ames, IA 50010, USA; 4Department of Pathobiology, College of Veterinary Medicine, Auburn University, Auburn, AL 36849, USA; smf0076@auburn.edu

**Keywords:** cell-mediated immunity, vaccine interactions, modified live vaccination, RB51, BVDV

## Abstract

Cell-mediated immunity is an important component of the immune response for intracellular pathogens. Live vaccines containing different pathogens are used concurrently in the field but are generally licensed individually. Concurrent administration of these vaccines has led to concerns about immune interference. The goal of this study was to characterize BVDV-specific responses to vaccination and determine the effect of concurrent *Brucella abortus* strain RB51 (RB51) vaccine administration. Peripheral blood mononuclear cells (PBMCs) from cattle vaccinated with a modified-live viral (MLV) vaccine containing BVDV, both RB51 and an MLV, or unvaccinated controls were utilized to evaluate antibody titers and the frequency of interferon gamma (IFN-γ) production within CD4^+^, CD8^+^ T cells, and NK cell populations via flow-cytometry. Our data demonstrated the lack of vaccine interference following concurrent administration of two common bovine MLVs and may even suggest some level of enhanced IFN-γ production with concurrent administration.

## 1. Introduction

Bovine viral diarrhea virus (BVDV), a member of the Pestivirus genus (Flaviviridae family), is a viral pathogen of cattle that has been implicated in the modulation of immune responses characterized by immunosuppression. This immunosuppression is dictated by alterations in immune cell function, namely, number and immunophenotype [1]. Numerous commercially available vaccines contain BVDV strains either alone or in combination with other bovine pathogens. Furthermore, concurrent administration of bovine vaccines against BVDV and other reproductive pathogens, such as *Brucella abortus*, is often co-administered pre-breeding. Concerns regarding vaccine or immune interference [2] imposed on cattle and other species is a topic of concern. However, due to the inability to test all vaccine combinations during licensure, most vaccines, unless licensed as combination vaccines, are not tested concurrently, and the effects of concurrent vaccination on immune responses or clinical responses are rarely evaluated or identified.

Following BVDV vaccination, titers have historically been used as the primary means of evaluating responses to vaccination but establishing a correlate of protection via titer value is not well established [3]. However, cell-mediated immunity (CMI) induced by vaccination is considered important in protection against BVDV and *Brucella* infection [4] and in inducing fetal protection. These responses, characterized by increased interferon gamma (IFN-γ)-producing T helper 1 (T_H_1) cells, mediate control and clearance of these pathogens [5,6,7,8].

Previously, we evaluated the effects of co-administering a viral modified live vaccine (MLV) and *B. abortus* strain RB51 (RB51), the commercial vaccine against bovine brucellosis, on the immune response to RB51. In that work, we demonstrated that the immune response to RB51 was not affected by the presence of an MLV [9]. The aim of this study was to evaluate whether RB51 vaccination had an effect on BVDV-specific immune responses following concurrent vaccine administration.

## 2. Materials and Methods

### 2.1. Animal Vaccination and Sampling

Hereford cross heifers, 4–6 months of age, were purchased and housed outdoors at the National Animal Disease Center (NADC) campus in Ames, Iowa. On arrival at the NADC campus, all animals were dewormed with Cydectin pour-on (Elanco, Greenfield, IN, USA), vaccinated with Ultrabac 8 (Zoetis, Kalamazoo, MI, USA), and treated with Draxxin prophylactically (Zoetis). All animals were healthy and in good body condition. Following acclimation for 6 weeks, cattle were randomly assigned to the following treatment groups: unvaccinated controls (Control, n = 6), MLV (Bovishield Gold 5) single vaccinates (MLV, n = 6), and MLV and RB51 (Colorado Serum, Denver, CO, USA) co-vaccinates (Combo, n = 6). All vaccines were administered according to the manufacturer’s recommendations. Where indicated, a subset of animals was revaccinated with the MLV 18 weeks post-initial vaccination, according to the manufacturer’s recommendations. Blood samples from all animals were taken at 18 weeks post-initial vaccination and at 4 weeks post-re-vaccination to evaluate CMI and humoral responses to the MLV vaccine. This study was carried out in strict accordance with the Guide for the Care and Use of Laboratory Animals of the National Institutes of Health (NIH) and the Guide for the Care and Use of Agricultural Animals in Agricultural Research and Teaching of the Federation of Animal Science Societies. The NADC Institutional Animal Care and Use Committee (IACUC) approved protocols prior to implementation.

### 2.2. BVDV Strains

Non-cytopathic (ncp) field strains, BVDV-1a (PI34) and BVDV-2a (PI28), were selected to stimulate peripheral blood mononuclear cells (PBMC) and evaluate BVDV-specific cell-mediated responses. Cytopathic reference strains BVDV-1a (Singer) and BVDV-2a (296c) were used for virus neutralization assays. Details regarding complete genome sequencing and BVDV isolate characterization are previously described in the literature, in addition to GenBank accession numbers [10,11]. BVDV strains were propagated in Madin–Darby bovine kidney cells (MDBK) that had been tested and were free of BVDV and HoBi-like viruses, as previously described and according to standard protocol [12].

### 2.3. Virus Neutralization Titers (VNT) to BVDV

Blood was collected into serum separator tubes (BD Vacutainer^®^, Becton Dickinson, Franklin Lakes, NJ, USA) and serum was obtained by centrifugation at 800× *g* for 30 min. Samples were stored at −20 °C until analysis. VNTs were determined using cytopathic isolates BVDV-1a (Singer) and BVDV-2a (296c), as previously described [5]. Results were expressed as the reciprocal (Log2 base) of the highest serum dilution able to inhibit the appearance of CPE in cells. Antibody titers equal to or higher than 2.5 log2 were considered positive for all samples. Serum samples were run in 96-well plates with 3 wells per dilution, and titers were calculated by the Spearman–Karber method.

### 2.4. Evaluation of CMI Responses to BVDV Vaccination via PrimeFlow RNA Assay

Blood was collected into tubes containing 2× acid citrate dextrose (ACD) for isolation of peripheral blood mononuclear cells (PBMCs), as previously described [9]. The evaluation of BVDV-specific cell-mediated responses was performed, as previously described [5]. Briefly, 1 × 10^6^ PBMC were plated onto round-bottom 96-well plates in duplicates for each stimulation condition. Approximately 24 h after plating, PBMC were left unstimulated or stimulated with BVDV-2a (PI28) at a multiplicity of infection (MOI) of 1 and maintained at 37˚ C with 5% CO_2_ for an additional 24 h. PBMCs were then harvested and stained for surface expression of CD4, CD8, and CD335, as well as intracellular IFN-γ and BVDV mRNA, as previously described [5]. Cells were then analyzed via flow cytometry using a FACS Symphony A5 (BD Bioscience, San Diego, CA, USA), and flow cytometry data were analyzed using FlowJo software (FlowJo, Ashland, OR, USA).

### 2.5. Statistical Analysis

Statistical analyses were performed using a one-way ANOVA with multiple comparisons and Tukey’s correction to determine statistical differences between treatment groups at the time points analyzed. All statistical calculations and analyses were performed using GraphPad Prism 9 (GraphPad Software, San Diego, CA, USA). Means were considered significantly different at *p* < 0.05.

## 3. Results

### 3.1. BVDV-Specific Virus Neutralizing Titers (VNT)

All calves were seronegative for BVDV-1 and BVDV-2 prior to vaccination. The non-vaccinated calves (Control) remained seronegative through 18 weeks post-vaccination. Vaccinated calves (MLV and Combo) seroconverted, and BVDV-1a VNT for MLV vaccinates ranged from 6.1 to 8.1 (mean = 7.3), while for the Combo vaccinates, VNT ranged from 6.8 to 8.5 (mean = 8.0). BVDV-2a VNT values ranged from 8.1 to 8.5 (mean = 8.4) for MLV vaccinates and from 8.1 to 10.8 (mean = 9.3) for Combo vaccinates. These data suggested that co-administration of MLV with RB51 did not affect the generation of neutralizing titers to BVDV.

### 3.2. Evaluation of BVDV-Specific Cell-Mediated Responses

Using an *in vitro* stimulation assay to measure BVDV-specific IFN-γ mRNA responses within CD4^+^, CD8^+,^ and CD335^+^ cells, we assessed T cell and NK cell-mediated responses following vaccination. At 18 WPV, both CD4^+^ (Figure 1A) and CD8^+^ (Figure 1B) T cells showed a trend increase (*p* > 0.05) in the frequency of IFN-γ mRNA positive cells in both MLV and Combo vaccinated groups compared to control animals. Similarly, we observed an increase in the frequency of IFN-γ mRNA-positive CD335^+^ NK cells in both MLV and Combo vaccinated groups compared to the control animals (Figure 1C). However, only the Combo vaccinated animals showed a statistically significant increase in IFN-γ mRNA-positive CD335^+^ cells compared to control animals (Figure 1C). Altogether, these data would suggest that co-administration of MVL and RB51 does not have a detrimental effect on the BVDV CMI response at the time point analyzed. Furthermore, following vaccination, regardless of the vaccination treatment, there are almost four times as many CD335^+^ cells (MLV mean = 1.94; Combo mean = 2.75) expressing IFN-γ mRNA as compared to CD4^+^ (MLV mean = 0.28; Combo mean = 0.45) and CD8^+^ (MLV mean = 0.51; Combo mean = 0.53) T cells. This raises interesting questions regarding the role of NK cells in the response to MLV vaccination.

### 3.3. Evaluation of BVDV-Specific Humoral and Cell-Mediated Responses Following MLV Re-Vaccination

In order to assess the amnestic response to BVDV in the MLV and Combo vaccinates, a subset of cattle was re-vaccinated with the MLV, and both humoral and cellular responses were reassessed. Following re-vaccination, BVDV-1a VNT values ranged from 5.5 to 8.1 (mean = 7.1) for MLV re-vaccinated and from 7.1 to 9.1 (mean = 8) for Combo re-vaccinated. BVDV-2a VNT values ranged from 8.1 to 11.1 (mean = 9.9) for MLV re-vaccinated and from 9.4 to 10.8 (mean =10.1) for Combo re-vaccinated. These data demonstrate that following a re-vaccination with MLV, the amnestic antibody response to the MLV remains comparable between the two vaccinated groups and suggests that the co-administration of the vaccines does not have an impact on antibody titers.

In general, and as expected, we observed an increased frequency of IFN-γ mRNA-positive cells in the MLV and Combo vaccinated groups compared to control animals following re-vaccination (Figure 2), and in higher overall frequencies compared to initial vaccination (Figure 1), as expected for an anamnestic response. However, in the Combo-vaccinated group, we observed a statistically significant increase in the frequency of CD4^+^ (Figure 2A), CD8^+^ (Figure 2B), T cells, and CD335^+^ (Figure 2C) cells compared to control animals. These data would suggest that Combo-vaccinated animals may have a slightly enhanced response compared to MLV-vaccinated animals. As observed following initial vaccination, the frequency of IFN-γ mRNA-positive cells within CD335^+^ cells appears to be almost four times in magnitude as compared to the CD4^+^ and CD8^+^ T cell compartments.

## 4. Discussion

We sought to describe and characterize potential vaccine interaction effects on the humoral and CMI responses when combining administration of a BVDV-containing MLV vaccine with RB51. Observations from this study were similar to the results from our previous work [9] and suggest that concurrent vaccine administration did not significantly impact BVDV-specific responses. We and others have extensively studied the humoral and CMI responses observed following BVDV infection [4,5,13,14], and the data presented here show a similar magnitude of both humoral and CMI responses to BVDV as seen previously. Concurrent RB51 and MLV vaccine administration did not reduce humoral and CMI responses to BVDV compared to MLV administration alone. Interestingly, however, upon re-vaccination with MLV, we observed an increase in the frequency of CD4^+^, CD8^+^, and CD335^+^ IFN-γ-producing cells in the Combo group compared to control animals. Collectively, the data provide evidence of a lack of vaccine interference and perhaps a potential benefit of concurrent RB51 vaccine administration upon re-vaccination. Furthermore, the data also highlight the potential role of NK cells in the response to vaccination.

These data are preliminary and part of a limited study, but they certainly support further evaluation of a possible role for RB51 in enhancing immune responses to other vaccine targets, such as BVDV, when administered concurrently. RB51 disseminates to non-draining lymph nodes and persists for up to 12 weeks following vaccination [15,16]. The dissemination and persistence of RB51 within lymph nodes could result in an altered inflammatory environment within lymphoid tissues that may enhance concurrent responses. It is worth noting that the effect of RB51 appears more evident upon re-vaccination and is primarily seen in the CMI response, as VNT did not differ between MLV and Combo vaccinates post-vaccination and post-re-vaccination.

Another interesting finding was the NK cell response observed following vaccination and re-vaccination. The frequency of IFN-γ mRNA-positive CD335^+^ cells was greater than that of CD4^+^ and CD8^+^ T cells following initial vaccination. Furthermore, upon re-vaccination, we observed an increase in the frequency of IFN-γ mRNA-positive CD335^+^ cells compared to the initial vaccination and to a greater magnitude than both CD4^+^ and CD8^+^ T cells. NK cells are one of the first cell types to respond in the face of vaccination or disease challenge [17], and cytokines from NK cells contribute to and may significantly help shape the adaptive response that follows [18]. NK cells can enhance vaccine-induced responses by producing IFN-γ and stimulating antigen-presenting cells [19,20]. Furthermore, it has been demonstrated that both vaccination and infection may shape the NK cell repertoire, sometimes resulting in subsets of NK cells with hyperfunctional states [21,22]. Targeting NK cells to enhance vaccine responses is an area of current research [23], and the data presented here point to a role for NK cells in the response to BVDV in cattle.

While limited in scope, the work does provide some interesting observations for further evaluation. Furthermore, the concurrent vaccine administration model utilized in these studies provides a framework for teasing apart and characterizing not only vaccine–vaccine interactions but also the possible role of NK cells in modulating vaccine-induced adaptive immune responses in cattle.

## Figures and Tables

**Figure 1 viruses-17-00553-f001:**
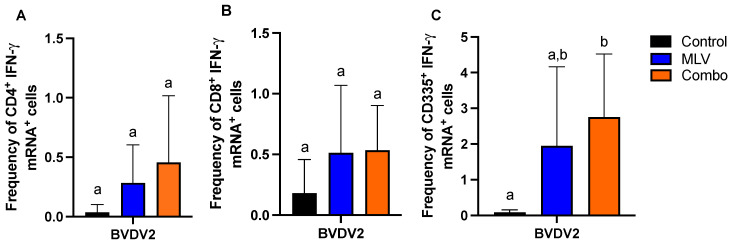
Frequency of IFN-γ mRNA-positive lymphocyte subsets within PBMC cultures in response to BVDV-2 stimulation at 18 weeks post-vaccination. Shown are the frequencies of (**A**) CD4^+^ T cells, (**B**) CD8^+^ T cells, and (**C**) CD335^+^ NK cells for control (black), modified live (MLV; blue), and Combo (RB51 and MLV; orange). Letters indicate statistical differences between groups (*p*-value ≤ 0.05). Error bars represent standard deviation.

**Figure 2 viruses-17-00553-f002:**
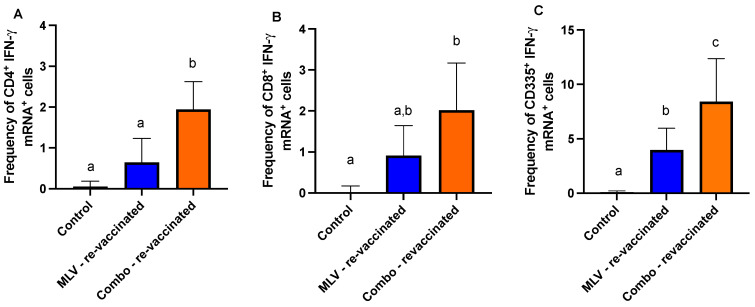
Frequency of IFN-γ^+^ mRNA lymphocyte subsets within PBMC in response to BVDV-2 stimulation. Shown are the frequencies of (**A**) CD4^+^, (**B**) CD8^+,^ and (**C**) CD335^+^ lymphocyte populations expressing IFN-γ mRNA for control (black), MLV vaccinates re-vaccinated with MLV (MLV re-vaccinated; blue), and Combo vaccinates re-vaccinated with MLV (Combo re-vaccinated; orange). Letters indicate statistical differences between groups (*p*-value ≤ 0.05). Error bars represent standard deviation.

## Data Availability

All data generated in this study are provided in this manuscript.

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
