# Peer review of "Effects of Concurrent Administration of BVDV Modified Live Viral Vaccine and RB51 on Immune Responses in Cattle"

_viruses, 2025, doi:10.3390/v17040553_

Round 1
Reviewer 1 Report
Comments and Suggestions for Authors
Crawford et al. investigate the immune effects of concurrently administering a modified live BVDV vaccine with the RB51 vaccine in cattle. The study examines both humoral immunity, measured by virus-neutralizing titers, and cell-mediated immunity, indicated by IFN-γ mRNA production in CD4⁺, CD8⁺, and CD335⁺ (NK) cells. The researchers compared groups of cattle that were unvaccinated, vaccinated with the BVDV MLV alone, or co-vaccinated with RB51 and the MLV. Evaluations were conducted at 18 weeks’ post-initial vaccination and after a subsequent revaccination. Their data suggest that co-administration does not impair BVDV-specific immune responses and may even boost NK cell-mediated IFN-γ production following revaccination. However, the presentation of the study's data does not fully support the authors' conclusions. The following comments outline the main issues:
- Some figures suffer from unclear axis labeling and overly large error bars, resulting in p-values greater than 0.5 in Figures 1A and 1B. This lack of clarity hampers the presentation and interpretation of data trends.
- The manuscript only provides basic information on cattle breed, age range, and housing conditions, without addressing their health status, immunization history, or genetic background. These factors may influence immune responses and compromise the accuracy and reproducibility of the results.
- Key experimental procedures, such as the virus neutralization assay and cellular immune detection, are not described in sufficient detail. For example, the specific components used for cell stimulation are not clearly explained.
- With only six cattle per group, the sample size is small, diminishing the robustness of the conclusions. Moreover, the study examines immune responses only at selected time points, failing to track long-term immunity following vaccination. This limits the evaluation of the sustained effects of the combined vaccination.
- The study relies solely on immunological markers without correlating these responses with clinical protection against BVDV infection, reducing the practical relevance of the findings.
- Only one type of BVDV MLV vaccine was used, which restricts the generalizability of the conclusions to other vaccine formulations.
- The analysis focuses primarily on IFN-γ as an indicator of cell-mediated immunity and does not assess other cytokines or functional parameters that could provide a broader understanding of the immune response.
Author Response
Reviewer 1
1. Some figures suffer from unclear axis labeling and overly large error bars, resulting in p-values greater than 0.5 in Figures 1A and 1B. This lack of clarity hampers the presentation and interpretation of data trends.
We thank the reviewer for their comment, but we are unsure as to what they are referring to with “unclear axis labeling.” All the axis are appropriately labeled. Unfortunately, there is nothing we can do about the large error bars: the data is the data. We appreciate that this comment is related to another comment by the reviewer (#6), regarding the number of animals used in the study. We expect to have animal to animal variability in the measured immune responses, which is a major contributor to the variation observed here.We understand the comment and do appreciate that the error bars are visually unappealing, but we are basing our conclusions on the statistical analysis.
2. The manuscript only provides basic information on cattle breed, age range, and housing conditions, without addressing their health status, immunization history, or genetic background. These factors may influence immune responses and compromise the accuracy and reproducibility of the results.
We thank the reviewer for the comment. We have added additional information on the processing of the cattle done upon arrival to our campus and their health status. As far as the genetic background, the information we obtained from the producer was that they were Hereford crosses. We did not run any additional tests to determine specific genetic background. However, we appreciate the comment, as it is known that genetic background can influence immunity. As it is now noted in the manuscript, all cattle were allowed to acclimate following arrival to our facility and they were all healthy and in good body condition, at the time of and throughout the study.
3. Key experimental procedures, such as the virus neutralization assay and cellular immune detection, are not described in sufficient detail. For example, the specific components used for cell stimulation are not clearly explained.
We thank the reviewer for the comment. However, all of this information is provided in the manuscript. As this is a brief communication, we are limited in space, therefore, we used citations to describe our work to ensure reproducibility; but we appreciate that the descriptions are limited in the text. As such, all the procedures described in this manuscript have been extensively described in other publications by our group, and we have provided the appropriate references. For the virus neutralization assay, information is provided in lines 75 - 82. As for the components used in cell stimulation, that is described in lines 84 – 88. We have provided some additional clarification, but again, are limited in space and rely on citations. Additionally, we did catch that the appropriate reference was not cited for the PMBC isolation procedure, and we have rectified that.
4. With only six cattle per group, the sample size is small, diminishing the robustness of the conclusions. Moreover, the study examines immune responses only at selected time points, failing to track long-term immunity following vaccination. This limits the evaluation of the sustained effects of the combined vaccination.
We very much appreciate the reviewer’s comment. We are aware that working with an outbred population always results in increased variability in immune responses. We agree with the reviewer that having a higher n would have increased the robustness of our conclusions and possibly made our error bars smaller. However, 6 animals per group is commonly used in studies, and while it may be a small sample size when you compare it to small laboratory animals, it is not a subpar number when it comes to large animals. Yes, our analysis of the immune response is limited, and yes, this limits the long-term evaluation of those responses, and we acknowledge that in the manuscript. Still, we believe the data is worth sharing, and it’s why it’s presented as a brief communication and not a full paper.
Additionally, in agreement with the reviewer, we are also interested in the long-term effects of vaccine co-administration on immune responses. However, such a study would require a longitudinal study to track such responses, extending at the very least over a 2-year period, if we are to appropriately assess the protectiveness of such responses, which includes assessment of protection to the developing fetus. This is also pointed out by the reviewer in the comment below (#5). The data presented here, in this brief communication, is an initial assessment to help inform if concurrent administration of these two vaccines, used to prevent fetal infections, causes unintentional effects that may impact protective response when used in heifers pre-breeding. Further work expanding on these findings is warranted but is outside the scope of this manuscript.
5. The study relies solely on immunological markers without correlating these responses with clinical protection against BVDV infection, reducing the practical relevance of the findings.
Again, we agree with the reviewer that protection data from challenge studies ultimately determine the success of the immune response. However, assessment of cellular mediated immunity and humoral responses are known correlates of protection for BVDV and commonly used to assess vaccine responses prior to doing more expensive challenge studies. The MLV used in this study is a commercial vaccine known to be efficacious against challenge, thus it does not require further proof of its protective effect. Had we observed that the addition of RB51 resulted in reduced responses to BVDV, then certainly a challenge study would have been necessary or warranted. Ultimately, the relevance of this brief communication is not in providing challenge data, but in providing evidence for a lack of vaccine interference, even with a limited analysis.
6. Only one type of BVDV MLV vaccine was used, which restricts the generalizability of the conclusions to other vaccine formulations.
Thank you again for the comment and absolutely agree. We only utilized one MLV vaccine, and could have included other commercial vaccines, including killed formulations. However, the MLV used in this study, based on market research, is one of the most widely used MLV vaccines on the market, especially for reproductive protection claims against BVDV. Given its wide use in the field, we sought to use it as it would be the vaccine most likely used concurrently with RB51 in reproductive age females. Our conclusions are solely based on the data provided for the MLV, and we do not make any generalizations regarding other vaccine formulations. While we agree with the reviewer that a larger study including other vaccines would be incredibly informational, this is outside the scope of this manuscript, but it is certainly something to consider for future work.
7. The analysis focuses primarily on IFN-γ as an indicator of cell-mediated immunity and does not assess other cytokines or functional parameters that could provide a broader understanding of the immune response.
We appreciate the reviewer’s comment, and yes, we do utilize IFN-γ as the only indicator of cell-mediated immunity. While a valid point, as stated in our introduction, IFN-γ is the primary cytokine produced by Th1 cells, which are required to confer protection against both BVDV and brucellosis. The work presented here is a brief communication to present a snapshot of the response to concurrent administrations of these vaccines. Our conclusions are solely based on the data, however limited in this case. We agree that looking at other indicators of cell mediated immunity would broaden our understanding of the immune response, but that is outside the limited scope of this brief communication.
We thank the reviewer for their comments and their time in reviewing our manuscript. We hope the reviewer finds our responses satisfactory, but welcome any additional comments and/or suggestions.
Reviewer 2 Report
Comments and Suggestions for Authors
This study focused on whether the combined vaccination of two kinds of vaccines would affect the respective immune efficacy (including cellular immunity, humoral immunity and immune duration, etc.) of the two vaccines in farm-produced cattle. The combined vaccination of MLV of BVDV and Brucella abortus RB51 was taken as an example. Two limited tests, measuring BVDV neutralizing antibodies and cellular immune response in immunized cattle, showed that the combination of the two vaccines (including booster immunization) had no effect on the humoral neutralizing antibodies of BVDV, and the initial combination of the two vaccines had little effect on the number of CD4+ and CD8+ T cells of BVDV. However, the number of CD335+ NK cells in BVDV increased by three times in both the initial combined immunization and enhanced combined immunization, and the number of CD4+ and CD8+ T cells in BVDV increased by nearly one fold after enhanced combined immunization (not specifically described in the paper). The combined application of BVDV vaccine and abortive brucellosis vaccine in cattle breeding has lifted the doubt that the immunization efficacy of BVDV vaccine will be affected.
Throughout the study, there are the following contents that need to be improved:
- The study should add to the part that examines the effective humoral antibodies and cellular immune effects of the other component of the combined immunization vaccine - abortive brucellosis vaccine. Only when the combined immunization is shown to have no effect on the immune efficacy of the other vaccine can the two vaccines be combined for immunization. Therefore, the trial design needs to increase the number of experimental groups immunized with Brucella abortus RB51 alone.
- To determine whether the cellular immune effect in addition to measuring the number of cells, it is also necessary to measure the expression of IFN-γ mRNA nucleic acid and IFN-γ protein.
- Do wild strains BVDV-1a (PI34) and BVDV-2a (PI28) belong to different serotypes or subtypes? If they are only different strains, only one of them will be used to measure antibodies; if they belong to different serotypes or subtypes, two strains should also be used to measure cell effects.
- 4. the statistical analysis did not cover all, a. after enhanced immunization, the cellular immunity level of combined immunization is different from that of single immunization; b. Generally speaking, the humoral immune antibody level after enhanced immunization is also increased compared with the initial immunization.
- The MLV is repeated at the end of the article; Figure 1B line numbers overlap with ordinates.
Author Response
1. The study should add to the part that examines the effective humoral antibodies and cellular immune effects of the other component of the combined immunization vaccine - abortive brucellosis vaccine. Only when the combined immunization is shown to have no effect on the immune efficacy of the other vaccine can the two vaccines be combined for immunization. Therefore, the trial design needs to increase the number of experimental groups immunized with Brucella abortus RB51 alone.
We appreciate the reviewer’s comment. For this study, we only have the given experimental groups, so unfortunately, we cannot add additional data. However, we do agree that to fully understand the effects of co-administration, we need to assess both sides of that response. We have previously published another manuscript where we assess the immune response to RB51 when co-administered with the BVDV MLV, which may provide the reviewer with a satisfactory response to their question.
Crawford, L., Falkenberg, S., Putz, E. J., Olsen, S., & Boggiatto, P. M. (2023). Effects of concurrent administration of modified live viral vaccines with RB51 on immune responses to RB51. Frontiers in Veterinary Science, 10, 1105485.
2. To determine whether the cellular immune effect in addition to measuring the number of cells, it is also necessary to measure the expression of IFN-γ mRNA nucleic acid and IFN-γ protein.
We thank the reviewer for the comment and very much appreciate that mRNA expression may not correlate to protein production. However, we have developed this specific assay to measure responses to BVDV, and we have extensively shown that this is a successful method of measuring such responses (Falkenberg et al. 2020). The reference for this assay is found below.
At this time, the only data we have from these studies is mRNA IFN-γ data and are not able to provide information regarding protein production. However, while we did not publish the preliminary data when developing the mRNA assay, we did look at IFN-g protein accumulation and did see the same trends between mRNA and protein for the CD4 population. During the initial development of the assay, we optimized the timing of the mRNA assay (24 h post-stimulation) to determine peak mRNA expression and determine differences in frequency of intracellular BVDV. The reason we ultimately decided to use the mRNA-based assay was to have the opportunity to assess IFN-g within the NK (CD335) population. If we leverage a traditional protein accumulation assay, we had to wait approximately 4 days after stimulation to assess recall responses and the NK population was not able to be assessed. As observed in the current study, the NK population is an important contributor of IFN-g that needs to be explored and characterized, and this assay provides that opportunity.
Falkenberg, S. M., Dassanayake, R. P., Neill, J. D., Walz, P. H., Casas, E., Ridpath, J. F., & Roth, J. (2020). Measuring CMI responses using the PrimeFlow RNA assay: A new method of evaluating BVDV vaccination response in cattle. Veterinary immunology and immunopathology, 221, 110024.
Additionally, RT-qPCR is used commonly to assess mRNA cytokine responses in a variety of species. The PrimeFlow assay mRNA assay adds an extra layer of visibility by allowing mRNA expression frequency in respective cell populations.
3. Do wild strains BVDV-1a (PI34) and BVDV-2a (PI28) belong to different serotypes or subtypes? If they are only different strains, only one of them will be used to measure antibodies; if they belong to different serotypes or subtypes, two strains should also be used to measure cell effects.
Thank you for the comment. BVDV-1 and BVDV-2 are characterized as different Pestivirus species (not serotypes or subtypes) due to their genetic variability, but antigenically, they are cross-reactive. The 5-way MLV vaccine used in this study contains both a BVDV-1a and 2a strains in the vaccine, but the BVDV-2a strain in the vaccine is immunodominant. This is evidenced by the responses observed in the VNT where BVDV-2a titers are greater than the BVDV-1a.
This same trend exists for the IFN-g data with the IFN-g expression being greater when stimulated with the BVDV-2a isolate. Given that we reported VNT data demonstrating the increased VNT against the BVDV-2a strain compared to the BVDV-1a strain, we felt that the most impactful IFN-g data was to report the responses to BVDV-2a. Given the limited space in the brief communication, it did not seem useful to report the same trends for the BVDV-1a data albeit at just lower frequencies of IFN-g.
4. The statistical analysis did not cover all, a. after enhanced immunization, the cellular immunity level of combined immunization is different from that of single immunization; b. Generally speaking, the humoral immune antibody level after enhanced immunization is also increased compared with the initial immunization.
We thank the reviewer for the comment. We agree with the reviewer that anamnestic responses, such as the ones induced by revaccination, would be greater in magnitude, as compared to primary responses to vaccination. That can be seen when looking at the frequencies (of all cell types) producing IFN-g. We appreciate that the reviewer noticed this difference. However, since the objective of the study was to determine the effects of RB51 on the responses to BVDV, we focused our statistical analysis within timepoints, and between the vaccinate groups, rather than over time. However, we have added a sentence to highlight this in the revaccination section.
5. The MLV is repeated at the end of the article; Figure 1B line numbers overlap with ordinates.
Thank you for the comment and we apologize for the oversight. We have made the requested corrections to the manuscript.
We hope the reviewer has found our responses satisfactory but welcome any additional suggestions or further explanation of our responses. Again, thank you for your time.
Reviewer 3 Report
Comments and Suggestions for Authors
The manuscript entitled "Effects of concurrent administration of BVDV modified live viral 2
vaccine and RB51 on immune responses in cattle" presents the results of analyses aimed at characterizing BVDV-specific response to vaccination and determining the effect of concurrent Brucella abortus strain RB51 (RB51) vaccine administration.
The studies described by the authors were conducted on 3 groups of animals: a control group - unvaccinated; a group in which only the BVD vaccine was used; and a group vaccinated against both BVD and brucellosis. The experiment used both neicypopathic field strains to stimulate peripheral blood mononuclear cells (PBMC) and assess BVDV-specific cellular responses, and reference cytopathic BVDV strains for virus neutralization tests.
The experiment demonstrated the lack of vaccine interference following concurrent administration of two common bovine MLVs and may even suggest some level of enhanced IFN-γ production with concurrent administration.
The present results supplement the earlier work published by the authors.
In the reviewer's opinion, the work is well written and is a valuable source of knowledge for veterinary practitioners. However, please read the text carefully again and remove unnecessary commas (e.g. line 54) or spaces (e.g. line 15).
Author Response
Reviewer comment: In the reviewer's opinion, the work is well written and is a valuable source of knowledge for veterinary practitioners. However, please read the text carefully again and remove unnecessary commas (e.g. line 54) or spaces (e.g. line 15).
Response: We thank the reviewer for their time in assessing our manuscript. As requested, we had reviewed the manuscript carefully and removed unnecessary commas and extra spaces. We apologize for the oversight.